# CFD Visualization in a Virtual Reality Environment Using Building Information Modeling Tools

**Jiayi Yan** * , **Karen Kensek, Kyle Konis and Douglas Noble**

School of Architecture, University of Southern California, Los Angeles, CA 90007, USA; kensek@usc.edu (K.K.); kkonis@usc.edu (K.K.); dnoble@usc.edu (D.N.)

* Correspondence: jiayiyan@usc.edu

**Abstract:** Scientific visualization has been an essential process in the engineering field, enabling the tracking of large-scale simulation data and providing intuitive and comprehendible graphs and models that display useful data. For computational fluid dynamics (CFD) data, the need for scientific visualization is even more important given the complicated spatial data structure and large quantities of data points characteristic of CFD data. To better take advantage of CFD results for buildings, the potential use of virtual reality (VR) techniques cannot be overlooked in the development of building projects. However, the workflow required to bring CFD simulation results to VR has not been streamlined. Building information modeling (BIM) as a lifecycle tool for buildings includes as much information as possible for further applications. To this end, this study brings CFD visualization to VR using BIM tools and reports the evaluation and analysis of the results.

**Keywords:** CFD simulation; BIM; scientific visualization; virtual reality

## 1. Introduction

Building energy simulation and analysis, including computational fluid dynamics (CFD) simulations, are important aspects in the lifecycle of a building. CFD is a branch of fluid mechanics where numerical methods are applied to solve and analyze complex problems involving fluid flows [1]. From an architectural perspective, the application of CFD simulation can be divided into two areas: indoor and outdoor simulations [1]. For indoor environments, the main use of CFD is to simulate the natural ventilation or mechanical ventilation, heat transfer in certain spaces; for outdoor environments, the wind flow (velocity, pressure, turbulence, and temperature) can be calculated. As the indoor thermal environment is highly related to occupants' health and building energy saving because of the booming environmental issues and the threat of COVID-19 this year [2], it is vital to address CFD simulation and visualization environment condition prediction [3].

The post-process of CFD simulation results is important for design or other decision-making procedures. However, stakeholders like owners, architects, facility managers of a project in the design phase are influenced because of the difficulty in understanding CFD simulation results [3]. Thus, proper scientific visualization makes the data talks more straightforward and graphically favorable.

Virtual reality environment is increasingly popular in the field of scientific visualization to solve the problems mentioned above, based on using computer graphics to express complex ideas and scientific concepts [4]. A VR technology can assist in the unambiguous display of data structures by providing a rich set of spatial and depth cues [5]. There have been studies working on scientific visualization in virtual reality environment, for example, Helbig et al. researched meteorological raw data that were incorporated for 3D visualization using CAVE (Cave Automatic Virtual Environment) [6]; and Hosokawa et al. worked on CFD visualization in VR using Oculus Rift for observation [7].

However, the data interoperability, the overall workflow, and the VR effects of these studies can be improved.

Building information modeling (BIM) is an integrated database of building components that combines building information throughout the building's lifecycle including 3D graphics, parametric modeling, and user-supplied data that creates the virtual design and construction model. BIM provides the opportunity for each discipline to talk to each other by offering exchangeable data format [8]. The three main characteristics of BIM are data-rich models, object-based tools, and high interoperability in the whole project's lifecycle [9]. Despite the BIM advantages and features, BIM tools have not been used to connect CFD results to VR much. Therefore, in order to optimize the workflow and scientific visualization effects from CFD to VR and improve the data interoperability, this study firstly presents a workflow on CFD visualization in a VR environment using BIM tools. Meanwhile, an evaluation method for the visualization effects is proposed. This workflow is brought to life through a case study of an indoor space.

## 2. Literature Review

### 2.1. CFD Visualization

CFD software has been commercially available since the early 1980s in the engineering community for applications such as turbomachinery, aerospace, combustion, and mechanical engineering [10]. CFD simulations often contain high-dimensional data in a 3D volume. The display of phenomena associated with this type of data may involve complex 3D structures [11].

Because of the data complexity, CFD simulation results have the same problem as other building energy simulation results. It often contains hundreds, if not thousands, of point coordinates and several parameters of each point along with their values recorded in numerical data spreadsheets, which is hard to understand by non-professionals [3,12]. There has been progress on CFD visualization. For example, in the non-commercial CFD simulation software OpenFOAM, the results can be visualized in the form of 2D graphs that are plotted along the XY axis. The results from OpenFOAM can be exported and imported into an open-source 2D/3D data visualization application called ParaView [13]. However, the 3D visualized effects are limited to still false color and the ability of ParaView to visualize and render the simulated area (like a room with furniture) is insufficient. It can be improved to map the CFD results with the actual simulated area for better decision-making for users during the design phase. Besides, existing visualization tools such as ParaView, Tecplot 360, or ANSYS Fluent can be good visualization tools. However, they are more engineering-oriented, lacking user-friendliness for those outside the domain [12].

### 2.2. VR for Scientific Visualization

Virtual reality is proposed to solve the problem mentioned above. Both the data visualization and environment rendering could be improved. With VR, it is possible for users to experience a sense of place comparable or even identical to the real world or a fully imaginary environment [14,15].

Two important parameters, immersion and presence, are useful for researchers, users to evaluate VR. "Immersion is related to the physical configuration of the user interface of the VR application," and "presence is a subjective concept, associated with the psychology of the user" [15]. There are three categories of immersive VR systems: fully immersive (VR with head-mounted device), semi-immersive (CAVE), and non-immersive (mobile device or desktop-based VR) [15]. As for presence, that is, the psychological status, is highly dependent on the judgment by users themselves. For example, if the VR environment with a sea view and the sound of the wave aiming at calming people down help people ease their anxiety greatly, it can be considered a fully presence VR [16]. Helbig et al. researched meteorological raw data that were incorporated for 3D visualization using symbols representing parameters, such as the temperature, precipitation, and velocity from a worldwide view, with the VR effects realized using CAVE [6], which was a semi-immersive VR.

However, CAVE is very expensive, and the versatility is not very high. Another research on semi-immersive VR was to post-process the CFD simulation data in Unity 3D and create a VR environment that can interact with users by Wii U Pro controller [12]. This research brings in insights on using game engines like Unity 3D for data post-process rendering and better user interaction. There are researchers worked on a fully immersive VR system by simulating the natural ventilation of a single room in OpenFOAM and post-processing in ParaView. The researchers exported vtk dataset in Unity 3D, which is a VR game engine, and programmed using the visualization VTK toolkit to obtain the CFD wind flow splines in a VR environment; the results were visualized using the Oculus Rift headset [7]. By using a VR headset, this research improved the level of immersion [14], but the workflow was still orienting to engineering visualization and the graphic rendering effect was not improved much.

In recent research, the concept of 'extended reality' (or XR) has been introduced as a holistic term for VR/AR (augmented reality)/MR (mixed reality) technologies [14]. AR and MR technology are implemented for scientific visualization as well as integrate the information from the actual physical world through spatial computing. For example, researchers have developed a human–building interaction (HBI) model that combines four components including wireless sensor data, a CFD analysis of human–computer interaction, and AR visualization with a head-mounted device. The human–building interaction model allows for efficient processing and data transfer [17]. This example is impressive with techniques such as gestures to interact with the real-world environment while recognizing the importance of visualizing CFD data. However, this AR visualization demonstration only focused on the surface plane rather than air ventilation flow visualization. In addition to a head-mounted AR device, Lin et al. introduced an approach to provide users the CFD visualization with intuitive interaction with the indoor environment based on client–server framework through mobile phone [3], which was a demonstration for non-immersive device CFD visualization. Moreover, Zhu et al. developed a prototype system that CFD visualization of indoor thermal environment simulation with HoloLensHMD in order to integrate an indoor office environment information, which was an application of MR technology combining VR and AR [18]. However, an MR device like HoloLensHMD (market price $3500) is more expensive than a high-performance VR device like HTC VIVE (market price $499–$799). The research mentioned the application of Revit, which is one of the typical BIM tools, but the entire workflow was not streamlined and not related much to BIM in terms of data interoperability and project lifecycle.

Nevertheless, previous researches have discussed the AR/MR advantages for CFD visualization and great progress has been achieved. AR/MR cannot bring a fully immersive environment experience based on simulation or imaginary design and highly rely on spatial environment information from the physical world, which limits the decision-making ideas during the architectural design phase.

*2.3. Using BIM Tools to Streamline the Workflow from CFD Simulation to VR Visualization*

BIM is often misunderstood as a specific brand of software or is confused with computer-aided design (CAD). In fact, it is neither a single software nor CAD. BIM is an integrated database of building components as discussed above in *Introduction* section. BIM can foster efficiency in a building's lifecycle divided into 10 phases: programming, conceptual design, detailed design, analysis, documentation, fabrication, construction 4D/5D, construction logistics, operation and maintenance, demolition, and renovation [19]. Building simulation and analysis is a part of the building's lifecycle during the architectural design phase.

There are several simulation tools, such as BIM tools, concerning energy simulation in buildings. Research on energy simulation programs for net-zero energy building (NZEB) has led to the development of 10 energy simulation software programs graded and interpreted by their performance including in terms of usability, accuracy, intelligence, process adaptability, and interoperability [20]. Moreover, quantitative benchmarks were established to compare the capability of this software from aspects such as the metric, comfort level and climate, passive strategies, and energy efficiency.

The results included diagrams showing the usability, intelligence, interoperability, process adaptability, and accuracy of the ten energy-simulation software. The interoperability of the studied programs is moderate, allowing for easy exchange of either the geometry or the output data [20]. Researchers studied the interoperability between a BIM-based architectural model and performance analysis programs (EnergyPlus, eQUEST, Ecotect, and IES) based on the gbXML protocol and concluded that users should select appropriate analysis programs considering the interoperability of the energy analysis programs [21]. In addition, a high-performance building design requires a BIM-based thermal analysis during the design [22]. BIM tools help make better decisions owing to the coordinated and consistent information provided by the building information models [5]. As a result, energy simulation programs with high interoperability enrich the information provided by the building information models, and BIM tools make energy simulations easier and more accurate.

A CFD simulation, as part of the analysis of a building, is beneficial to be carried in software with high interoperability. The simulation and analysis results obtained using CFD can help enrich the building information models. Applying BIM tools, such as Autodesk Revit, to a CFD simulation helps in better decision-making regarding the design [23]. For example, there was an approach using BIM-based CFD simulation for wind environment prediction and evaluation in the early design stage [24], but the visualization was not addressed in this research.

With the increasing interest in BIM, there has been research on visualizing thermal building simulation data using simulation tools, for example, Ecotect Analysis, Revit, and the 3D visualization tool by 3DVIA to generate AR effects for projects such as the Gunzo room [5]. In the prototype system that includes CFD visualization with HoloLensHMD by Zhu et al., the BIM model was created in Revit [18]. However, the reason why choosing the BIM tool over other modeling tools in the general workflow is not clear. Moreover, the Revit model was exported as stl file and loaded in Pointwise software as a converter, then the model could be imported into OpenFOAM for CFD simulation. After the simulation, the data flowed from OpenFOAM to ParaView and Unity 3D. Compared to the approaches by Berger et al. and Hosokawa et al., whether there is an optimized workflow should be discussed [7,12,18].

A streamlined BIM workflow should include geometry modeling in Autodesk Revit, simulation and visualization, and analysis in Autodesk Simulation CFD [23], avoiding as much data loss as possible during the "round-tripping" process [25]. If the CFD simulation software has better BIM support, the "design-to-analysis" workflow is more convenient and precise for users because of the interoperability between the geometry input software and the simulation software. Post-processing is also part of the interoperability.

Multiple output data formats allow for more post-process software to read and use. There is an exception as well. For example, in IES-VE MicroFlo, the data exchange process is within the software itself, and there is no possibility of using the simulation results for visualization in other visualization software. Currently, several types of research related to CFD visualization and virtual environment are quite engineering oriented [3,17–19]. The output post-process file formats like vtk are often exported to create the virtual environment. However, vtk format is a common file format designed to offer a consistent data representation scheme for various dataset types that contains hundreds and thousands of complex structures of different geometry topologies and data attributes [3], which is not acceptable for BIM tools as well. Other output data formats that allowed for higher data interoperability should be investigated. There is another study that claims to integrate CFD, AR, VR, and BIM, but the role of BIM in the study was pretty ambiguous [26].

Considering the application of BIM tools, however, there is minimal research on the visualization of energy simulation data in a VR environment, particularly using BIM tools [7].

### 2.4. Summary

Three main conclusions can be drawn from the critical nature of BIM interoperability, visualization of CFD simulation, and the potential of using VR in the BIM workflow analysis.

1. Interoperability is significant for ensuring a data-rich and data-accurate BIM model throughout the life cycle. High interoperability of the software used in the BIM allows for effective data exchange so that the building information can be used for energy simulation or other applications such as scheduling and maintenance. Thus, energy simulation or other applications enrich the information of the model.

2. The CFD results require a high level of visualization to be understood, interpreted, and explained. Especially for CFD data, which have complicated structures, multiple parameters should be applied with a more intuitive visualization method. VR has emerged as a good choice for CFD as an innovative graphical representation method.

3. The workflow of a BIM tool in the entire life cycle of a product is being gradually developed and updated with newer methods. VR is an innovative method of viewing and sometimes manipulating data. It can be incorporated in the workflow by itself as a design tool and/or as a method for visualizing simulation data (Figure 1).

design ⟶ simulation ⟶ visualization ⟶ virtual reality

**Figure 1.** Workflow from building information modeling (BIM) to simulation to virtual reality (VR).

This research offers clues on how to include CFD-VR visualization in the BIM platform using a more simplified and consistent workflow that considers more architectural design requirements rather than engineering-focused only. A new data exchange format with high interoperability for post-process has been implemented. In addition, a fully immersive VR environment with improved graphic effects will be presented.

## 3. Methodology

### 3.1. Overview of Proposed Method CFD Visualization in VR Environment Using BIM Tools

The proposed method includes four parts, the BIM geometry, CFD simulation, 3D visualization, and virtual reality visualization, to realize the CFD visualization in a VR environment (Figure 2).

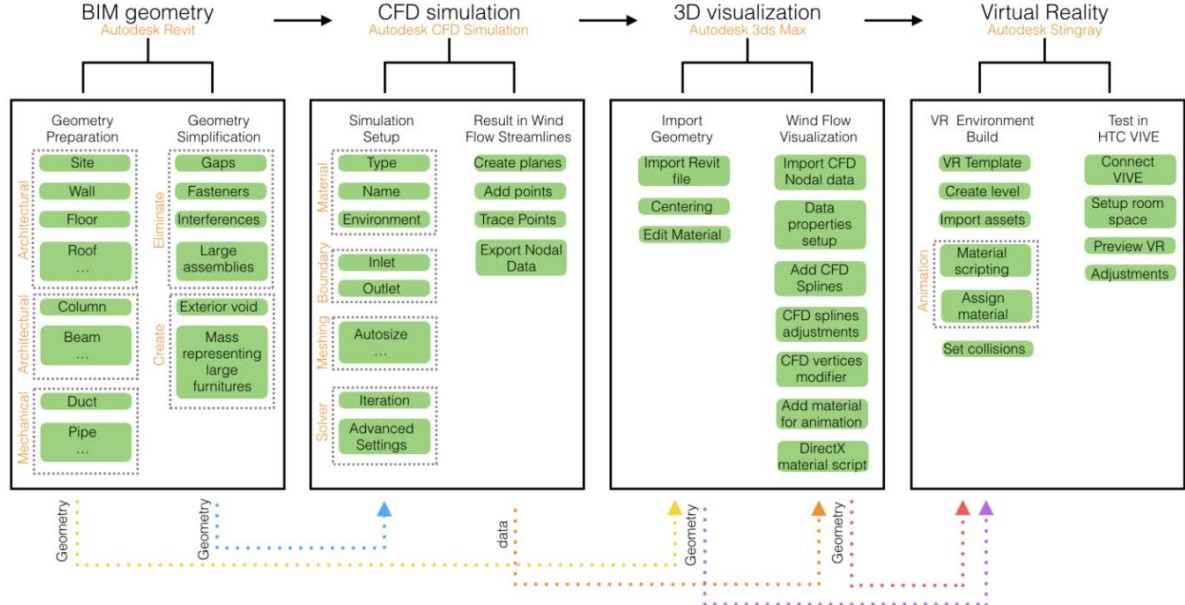

**Figure 2.** Proposed method of computational fluid dynamics (CFD) visualization in VR environment using BIM tools.

- BIM geometry. This part includes two sub-steps. First, a complete 3D model containing basic elements (all/one of architectural, engineering, and mechanical contents) was created. Particularly for the CFD simulation, in the second step, the model was simplified in advance to improve the CFD calculation speed and avoid unnecessary errors. For example, a cabinet can be simplified to a cuboid box without details.
- CFD simulation. After the modeling, a CFD simulation was carried out by importing the simplified 3D models from Revit. Before starting the simulation, the geometry was set in terms of the material used, boundary conditions, meshing, and solver requirements. These four setup tasks can be complicated or simplified depending on the simulation accuracy requirements of the user. If the simulation is processed successfully, the ingredient data for visualization are largely ready for export. The second step involved managing the data to trace the points from the inlet to the outlet of a certain plane while creating numerical data (nodal results) in the CSV format that can be used for the VR visualization.
- Three-dimensional visualization. In this step, the detailed BIM geometry was imported into Autodesk 3ds Max. Subsequently, the main step was to visualize the wind flow using the nodal results from the CFD simulation using the built-in CFD vertices modifier tool in Autodesk 3ds Max to editable splines that can be imported into the VR game engine. This step provides recognizable material for the later steps to create a VR environment.
- Virtual reality visualization. This step enables users to view the VR results using the HMD. The material created in the 3D visualization step was imported as an asset into Autodesk Stingray. The interactions and animations were made in Stingray. The test using HTC Vive HMD was also completed in the game engine as adjustments were required.

*3.2. Software Selection*

Software selection was important for overall efficiency and effects. As previously discussed in the *Literature Review* section, the software used for the methodology should have high interoperability in order to avoid data loss as much as possible. Besides, BIM tools should be included to guarantee the richness of the building information models and make the "round-tripping" design convenient among the four proposed phases.

- BIM geometry: Autodesk Revit was applied as the basis of the methodology. Revit has a great user interface and tutorials for architects and engineers to engage while the powerful modeling and information storage functions allow models of every kind to be built. The student version for Revit is available for research use.
- CFD simulation: as for simulation, although the representations of the results are adequate of all CFD simulation software, the priority was on interoperability. On one hand, Autodesk CFD simulation provides an add-in launcher in Revit, from which the model built in Revit can be imported to CFD simulation directly without an intermediate converter like pointwise [18]; on the other hand, Autodesk CFD simulation has multiple choices of output data formats including csv and nodal file that can be used in visualization programs, instead of vtk file, which facilitate data post-process. The simulation engine of Autodesk CFD is powerful and has well satisfying interior, exterior air-flow simulation, heat transfer simulation, etc. The user interface is friendly to both beginners and professionals. A student version is available.
- 3D visualization: the visualization of CFD results was of great importance since the software needs to read the CFD numerical data results at the same time to generate a graphically recognizable representation that can be used for the VR environment. 3ds Max works as the pre-process intermediate not only converting the Revit geometry into VR accepted geometry, it also has a significant CFD data modifier that can directly read nodal files from CFD simulation. A student version is available for 3ds Max.

- Virtual reality: the selection of game engine considered interoperability at first as well since the mainstream game engines on the market are all free to use and have powerful functions. Stingray can talk to 3ds Max directly because of the built-in tab of Stingray in 3ds Max sending geometry and materials back and forth or linking the files between 3ds Max and Stingray. In addition, visual programming is available in Stingray. A student version is available.

Overall, the software used for the proposed methodology are from the Autodesk series including Revit, CFD, 3ds Max, and Stingray, which, to some extent, facilitate the data exchange and function compatibility.

### 3.3. Data Flow

The datasets generated from different phase flow through different phases (Figure 2):

In the BIM geometry phase, two BIM model datasets are created: a detailed model and a simplified model. The detailed BIM model is a center hub for a project that includes all the information from different disciplines and project stages. In this research, the detailed BIM model goes to 3D visualization in 3ds Max and the VR environment in Stingray. The detailed geometry is important as it provides information of the simulated area, which can be the digital twin of a real physical world or the design in the architectural phase that makes that CFD data recognizable and accessible in a VR environment. The geometry is simplified only for the convenience of CFD simulation. Therefore, the simplified geometry data only goes to the CFD simulation phase. The detailed BIM geometry does not need to be loaded in CFD and can be directly loaded in 3ds Max.

Nodal file is exported from CFD as a simulation result that contains the point cloud data. It is the key to visualization. However, the nodal file is only an intermediate material for the visualization that only goes to the 3D visualization phase in 3ds Max. The pre-process of nodal file and detailed BIM geometry is a must before all the datasets go to the VR environment.

As the detailed BIM geometry and pre-processed CFD visualization datasets are loaded and matched in 3ds Max, they are jointed as a scene that is ready for game engine rendering in Stingray.

### 3.4. Evaluation

The visualizations in different software of the visualization phases have a different emphasis. As for the visualization in Autodesk CFD, it contributes more to the scientific application and the analysis of the data and has graphic deficiency as shown in existing workflows. The visualization in 3ds Max is convenient and graphically sound; however, it is not sufficiently informative in representing the meaning of the data without color legends. In Stingray, the VR environment with the CFD data brings a fully immersive experience with a relatively high presence. The visualizations in CFD and 3ds Max stages can be adopted separately depending on a user's requirements such as data analysis or representation; however, to realize CFD visualization in a VR environment, all three stages are necessary.

The user feedback testing method is adapted after exploring the three kinds of visualized CFD data [3]. Previous research has been worked on eight parameters to evaluate visualizations including the visualization in a VR environment [27]. Accuracy, Experience, and Graphics are selected in this work for the user feedback testing. The invited users are supposed to choose from "poor," "good," and "excellent" for the three criteria (accuracy, graphics, experience) for the visualizations in CFD, 3ds Max, and Stingray (VR) separately. An overall evaluation of the workflow is given in form of a radar chart to illustrate the dominant rating grade of all invited users.

- Accuracy: Accuracy refers to whether the graphics accurately show the data received from the CFD simulation;
- Graphics: The graphics include the rendering of the visualization and the possible visualization methods (different symbols);

- Experience: The experience indicates whether the visualization allows users to communicate with the visualization, such as for data analysis or to interact with the visualization.

## 4. Implementation

To test and demonstrate the feasibility and effectiveness of the proposed CFD VR visualization method, kitchen space was created as an implementation. A kitchen space requires air circulation, because the daily routines of cooking and dining may affect the environment of the kitchen. In addition, in a kitchen, there are often large assemblies such as cabinets and partitions that may influence the airflow. The size of the kitchen space should be appropriate for walking around in a VR case study.

### 4.1. Case Study Overview

The kitchen space was modeled after the kitchen design of a single-family house located in Los Angeles. The dimension was 42′ × 16′, 672 square foot in total. The kitchen space was divided into three separate areas with two partitions in order to create some variations in the wind flow (Figure 3).

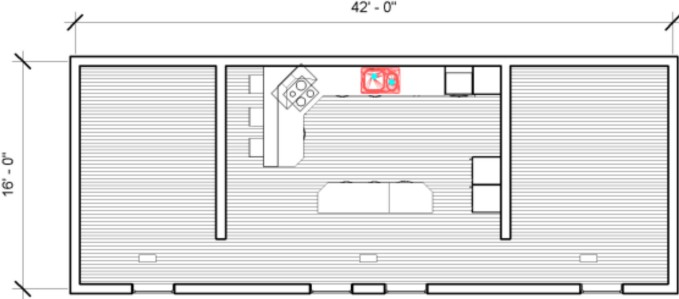

**Figure 3.** Floor plan of a kitchen space.

### 4.2. BIM Geometry

In Autodesk Revit 2016, a new project with an architectural template was built to start the project. Starting from the architectural elements, the wall, floor, roof, window, door, and furniture were placed in the Level 1 floor plan, and the mechanical, electrical, and plumbing (MEP) elements were included as inlet and outlet ducts (Figure 4). This is the detailed model for later geometry simplification and VR display (Figure 5).

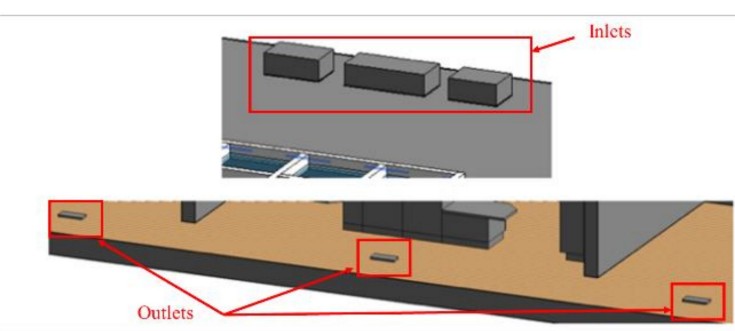

**Figure 4.** Inlets and outlets of the kitchen space.

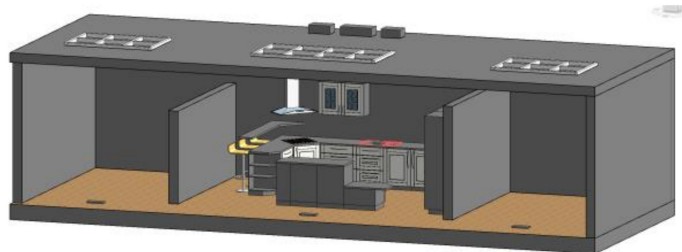

**Figure 5.** Three-dimensional perspective of the kitchen.

After the detailed model was built, the model was tailored to decrease the unnecessary surfaces for the simulation. Detailed furniture was replaced in the form of box-like masses with the same dimensions. The wind direction in the kitchen was from the inlet ducts to the outlet ducts (from the roof to the floor). As the model was prepared for a wind flow simulation, the window glasses will not influence the flow. Therefore, all the windows and skylights were eliminated from the model (Figure 6).

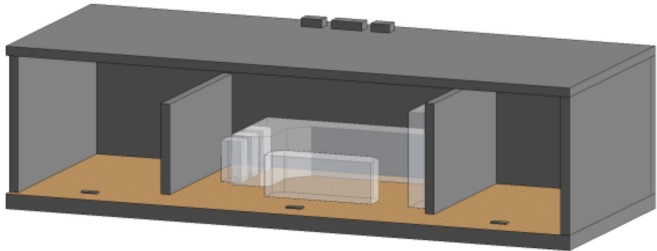

**Figure 6.** Simplified Revit model ready for CFD simulation.

### 4.3. CFD Simulation in Autodesk CFD

In the simplified kitchen_cfd.rvt file, the Launch Active Model icon was used to start Autodesk CFD 2016 for the simulation. In the Design Study Manager dialog, the file was saved in the appropriate folder with a project name.

### 4.3.1. Set Material

The indoor air volume was assigned Fluid type with the material name of Air; the floor was assigned as Solid, Timber Plank. Objects other than these two were all assigned as Solid, Gypsum Board.

### 4.3.2. Set Boundary

After setting up the material, setting the boundary conditions were the second task. As the kitchen is designed to let the air circulate from the roof inlet ducts to the outlet ducts on the floor, the inlet surfaces were assigned as *Velocity* type, with a speed of 85 in/s. The outlet surfaces on the floor are assigned to *Pressure* type, with 0 *psi*.

### 4.3.3. Mesh Sizing

In order to generate the nodes for calculation, the model was sized. In this case study, the Autosize function was applied for convenience and accuracy. After sizing, the Visual Style to Outline was changed to view the result.

### 4.3.4. Solver

After all the setup tasks are finished, the model was ready for solving. In the Solve Quick Edit Dialog, set the Iterations to Run of 20, as a balance of the computing speed and accuracy for a relatively small space. The convergence plot showed the progress of the calculation.

### 4.3.5. Result Viewer

In order to export the nodal data file for VR visualization, the airflow streamlines were created in CFD first. First, a plane was added in the middle of the space so that it covers the main points within the space. Then, the Trace button was used to add a rectangular grid onto the plane. In the Trace Quick Edit Dialog, the Add trace set tool generate the splines that traced each point from inlet to outlet.

### 4.4. Visualization in Autodesk 3ds Max

With the geometry from Revit and the nodal results from CFD, the 3D visualization could be realized in 3ds Max. In 3ds Max, a new project was started and saved.

### 4.4.1. Import CFD Data and Revit Model

In Revit, kitchen_detailed.rvt file was opened and exported the detailed kitchen model as FBX file that can be read by 3ds Max. Then, the model shows in the scene. The nodal file was imported as cloud points (Figure 7). The geometry and cloud points were overlapped seamlessly by centering to the origin (0, 0, 0).

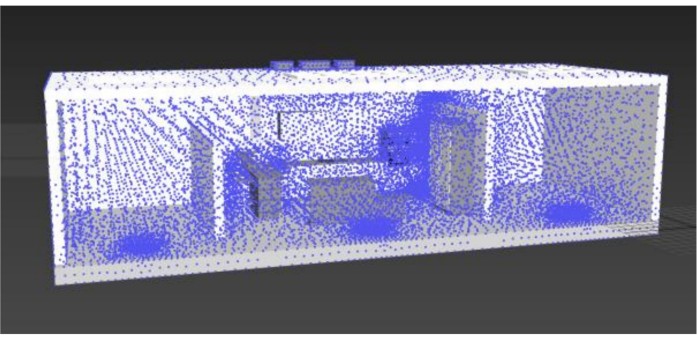

**Figure 7.** Import BIM geometry from Revit and CFD nodal results from Autodesk CFD.

### 4.4.2. Create CFD Streamlines

The cloud points were converted to editable poly and displayed as a box, and the kitchen geometry was hidden to better observe the splines. Using the tool to create CFD splines proposed in Chapter 3, the splines were generated with planes as the source and CFD data as the tracing points. The Number of Segments was 10,000, the Number of Examples was 30, and the Number of Steps was 4. By hiding the building geometry, the splines were created at the correct place in Autodesk CFD (Figure 8).

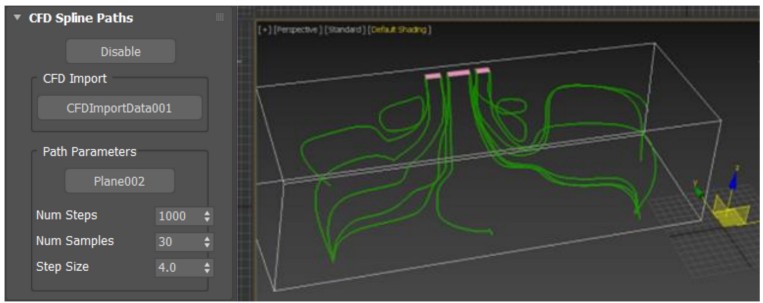

**Figure 8.** Creating the splines.

### 4.4.3. Add CFD Vertex Color Modifier

The splines were checked with the properties of Enable in Renderer and Enable In Viewport and Generate Mapping Coords. The thickness of the splines was set to 0.2′ for better visualization. In the Object Properties window, the Vertex Channel Display was checked as well. After the properties,

the vertex color modifier was added on top of the editable splines to generate the vertex color that represents the level of velocity. To boost the color, the Red Amount was increased from default 100 to 500 to enlarge the overall color range of the displayed data (Figure 9).

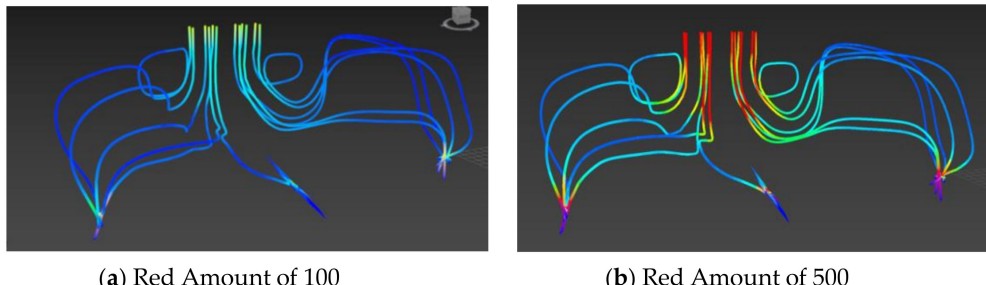

(**a**) Red Amount of 100　　　　　　　　　　　　　(**b**) Red Amount of 500

**Figure 9.** Red Amount of (**a**) 100 vs. (**b**) 500.

To avoid confusion when importing objects to Stingray, the splines and all the other objects were selected separately and then exported as FBX files. There were two FBX files: one for the splines and another for all the kitchen geometries.

*4.5. Virtual Reality Visualization*

On opening Autodesk Stingray, VR HTC Vive was selected as the template. All the kitchen geometries and CFD splines were imported as FBX files into Stingray.

To generate a colorful wind flow animation for the splines, a material script was created in 3ds Max. With this script imported into Stingray, a Stingray material was created with the parameters. The splines were selected and assigned the material named CFD animation material. The material was assigned a color map of an arrow, which created the animation of the wind flow from the inlet to the outlet. The special opacity parameter is assembled in Step 1, Step 2, and Step 3 (see Appendix A)

- Step 1: In this step, a rotation gizmo (rotator) is added to place and tilt the color map like an arrow or a gradient band based on the coordinates divided on the splines. It is a single vector node, which means the color map can be tilted only in one direction;
- Step 2: A panner is added to create a flowing effect. Using a panner is the same as creating the sea wave effect. The tiling node controls the intensity of the color map; the higher the value, the denser the color map placed onto the splines. Changing the speed U and speed V node will help increase or decrease the flow rate from the U and V directions of the splines. The time node is to inform the user and the program that the speed is modifiable;
- Step 3: The color map parameter is added and connected to step 1 and step 2 so that the Rotator, Tiling, Speed U, and Speed V can be controlled (turned on/off) by the opacity parameter.

Combining the three steps and connecting the opacity, metallic, and roughness to the standard base node, we can control the material script in the Stingray user's interface after the material script is imported into Stingray.

**5. Results**

A kitchen space indoor environment was adopted as an implement for the proposed method. The kitchen space was created with Revit and the model was loaded into CFD simulation with the CFD add-in launcher in Revit. CFD simulation was conducted and the nodal file was exported from CFD. With the interoperability between Revit and 3ds Max, the nodal file and the kitchen space geometry were imported into 3ds Max for pre-process. Finally, the pre-processed kitchen geometry and pre-visualized CFD simulation results were sent to Stingray for the VR environment. Users can experience with HTC VIVE HMD. As the method has proposed, the CFD visualization was presented in three software in the three phases of the method workflow, respectively.

## 5.1. CFD Visualization

CFD simulation result shows the particle tracing lines result, which uses the plane as a source to trace the path of each point on the plane, from inlet to outlet in the simulated space (Figure 10). The false color level shows the intensity of the particle. Besides, nodal file and csv file were exported from CFD (Figure 11).

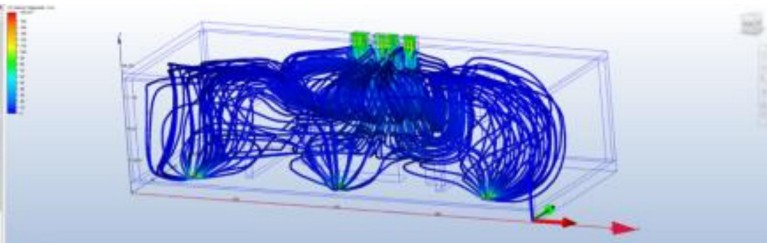

**Figure 10.** The wind flow tracing splines are added.

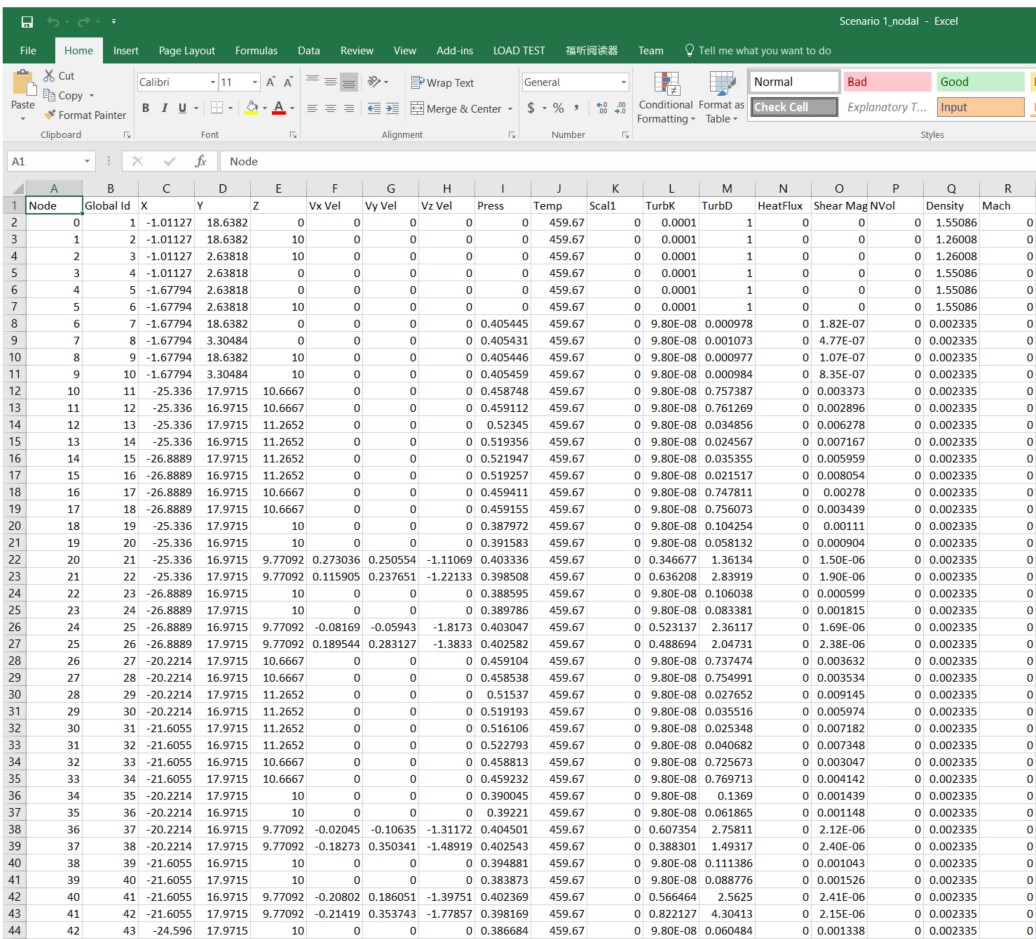

**Figure 11.** Nodal results from Autodesk CFD 2016.

## 5.2. 3ds Max Visualization

3ds Max is an important intermediate between CFD and VR environment. On one hand, 3ds max visualizes the CFD simulation result nodal file and create the airflow graphics from thousands of point clouds; on the other hand, the kitchen space geometry is brought in from Revit and is mapped with the visualized nodal file. It is of great importance to combine the simulation data and the

environment. What is more, the parameter called *Red Value* is able to change the color scheme, thus to create better visualization.

The final result was shown appropriately with kitchen space geometry (Figure 12).

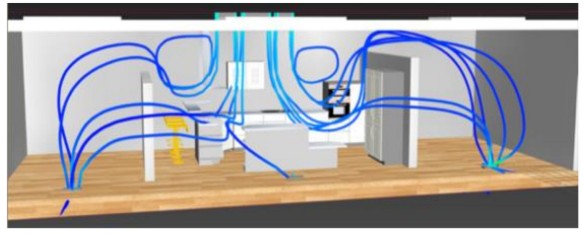
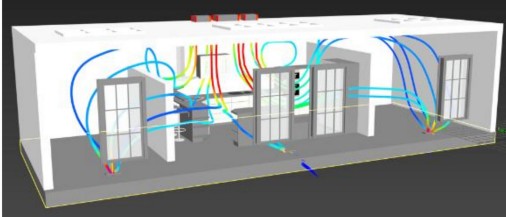

(**a**) Red Amount of 100        (**b**) Red Amount of 500

**Figure 12.** Red Amount of (**a**) 100 vs. (**b**) 500.

*5.3. Stinray Visualization*

In Stingray, the airflow can be rendered for a virtual reality environment. With the addition of the black and white gradient image in the color map, the wind flow animation is made thicker and now more closely resembles the flow of actual wind, which brought more animated diversity of the visualized data. The animation of CFD visualization was firstly discussed in this research. Although both the arrow image and the black and white gradient image are grayscale, the airflows are still coated with the visualized "CFD color." This is because when scripting the CFD animation material, a vertex color was used to represent the CFD velocity level; in this case, only the pattern of the image (colormap) is read regardless of the image (colormap) used. If no colormap is used, the airflow will be still instead of the flowing animated effect (Figures 13 and 14). Moreover, if the CFD splines in 3ds Max have increased Red Amount and then imported into Stingray, the splines in Stingray would be redder (Figure 15).

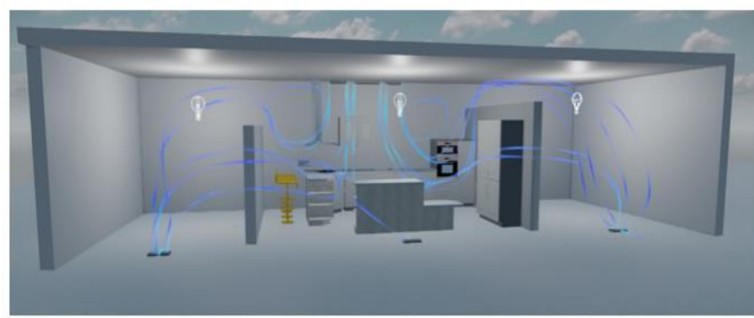
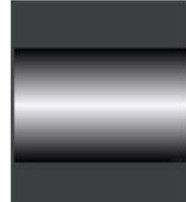

*Results after applying the Direct X shader material with gradient black and white band colormap*

**Figure 13.** Airflow animation with gradient color (**left**); the gradient color image (**right**).

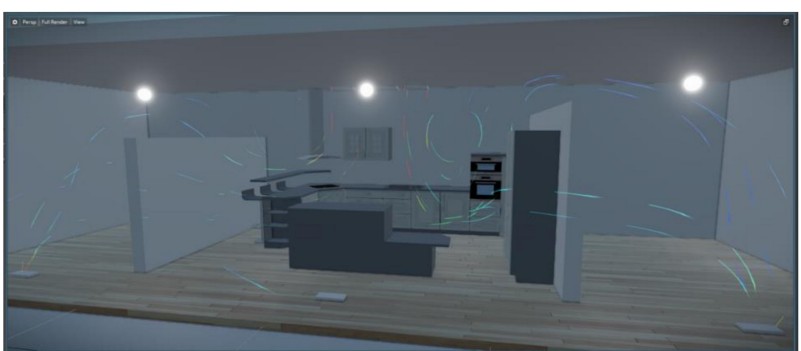
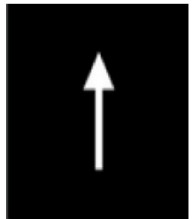

*Results after applying the Direct X shader material with black and white arrow*

**Figure 14.** Wind flow animation with arrow (**left**); the arrow colormap (**right**).

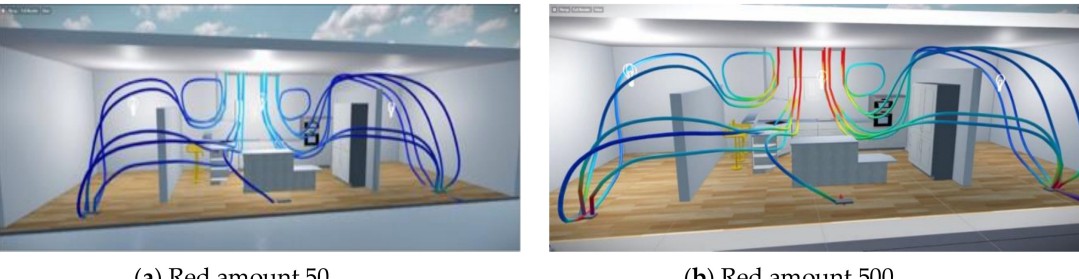

(**a**) Red amount 50       (**b**) Red amount 500

**Figure 15.** Airflows without animation in Stingray with red amounts of (**a**) 50 vs. (**b**) 500. (**a**) Airflows without animation in Stingray with red amounts of 50; (**b**) Airflows without animation in Stingray with red amounts of 500.

### 5.4. HTC VIVE Application

To drive VR equipment, a more powerful computer is required compared with the daily used ones, particularly in terms of the graphics card. Therefore, we employed a desktop with an ASUS Strix GTX 1080 8 GB Advanced Edition Gaming Graphics Card and an Intel Core i7-7700 K 4.2 GHz quad-core processor (with speed and memory) to power the VR equipment and create the project.

After the results were displayed in Stingray, the VIVE equipment was set up including sensors, joysticks, and HMD, and then the Play button was hit to start the testing. Users were able to walk around with the joystick and point to the position they wished to go and experience the CFD wind flow with the animations in the kitchen space (Figure 16).

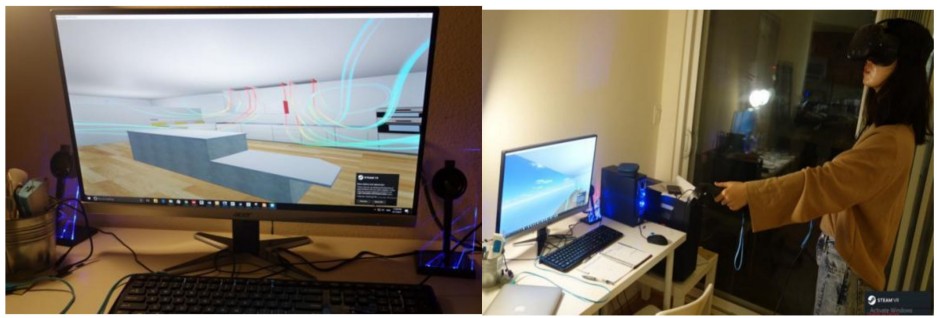

**Figure 16.** Test using HTC VIVE.

### 5.5. Users Feedback Test

To further validate the visualization effects in these three software, 8 random users were invited to give comments based on their opinions after they saw the visualized results or experiences using the HTC VIVE VR HMD. Among all 8 users, 2 of them were with a civil engineering background, 2 of them were in computer science background, 1 of them was with an architecture background, 1 of them was with MEP background, and the rest of the 2 people were in finance and education background separately. None of the users knew CFD before they were involved with the test. Therefore, the researcher explained the proposed method and briefly introduced the three software. 3 out of 8 of the users had VR HMD experience previously.

The interview results of the 8 users for CFD, 3ds Max, and Stingray (VR) were summarized separately (Tables 1–3). The concluded user rating was illustrated in Figure 17, demonstrating the dominant feedback of all the invited users that took the test. In general, over half of users (5 out of 8) agreed that the accuracy of CFD visualization was excellent; the rating for 3ds Max visualization was fairly on average; the visualization in Stingray VR environment brought excellent experience and good graphics effects.

**Table 1.** Summarized number of users' rating on CFD visualization (Unit: person/people).

| Criterion<br>Rating | Accuracy | Experience | Graphics |
|---|---|---|---|
| Excellent | **5** | 2 | 0 |
| Good | 3 | **4** | 1 |
| Poor | 0 | 2 | **7** |

**Table 2.** Summarized number of users' rating on 3ds Max (Unit: person/people).

| Criterion<br>Rating | Accuracy | Experience | Graphics |
|---|---|---|---|
| Excellent | 1 | 2 | 0 |
| Good | **5** | **4** | 1 |
| Poor | 2 | 2 | **7** |

**Table 3.** Summarized number of users' rating on Stingray (VR) (Unit: person/people).

| Criterion<br>Rating | Accuracy | Experience | Graphics |
|---|---|---|---|
| Excellent | 1 | **8** | 1 |
| Good | **7** | 0 | **7** |
| Poor | 0 | 0 | **0** |

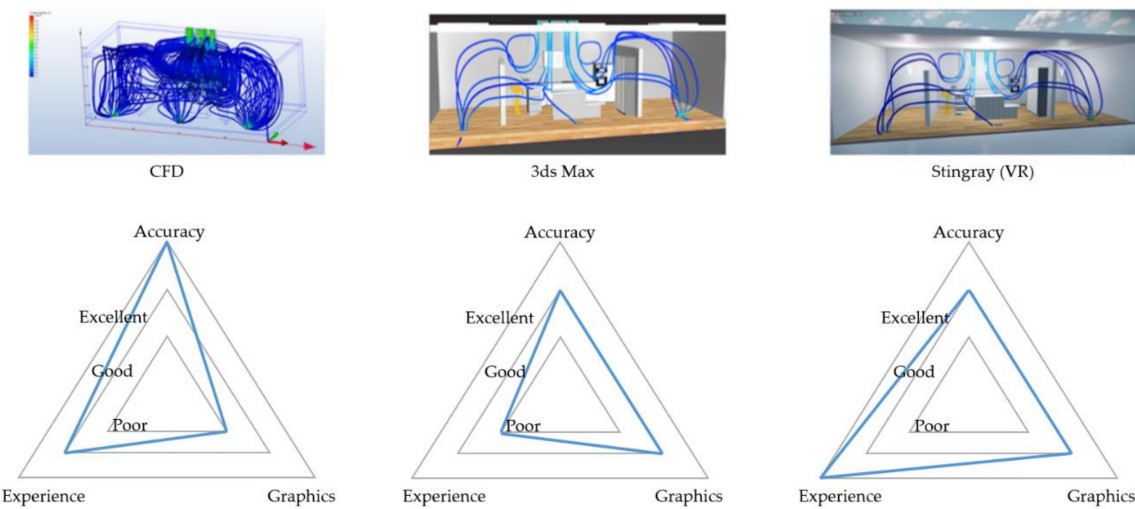

**Figure 17.** Rating of visualization results in Autodesk CFD, Autodesk 3ds Max, and Autodesk Stingray.

Besides, based on the short conversation with users, all of the users were astonished by the VR environment with CFD results. One of the users commented, "It is very easy for me to understand what the colorful flow represents and what is CFD." Although, there was a user indicating that if it is not for fun, the result from CFD software may be enough for scientific research.

## 6. Discussion and Future Work

There are several topics that require further studies from the aspects of BIM, CFD simulation, 3D visualization, and VR (Figure 18).

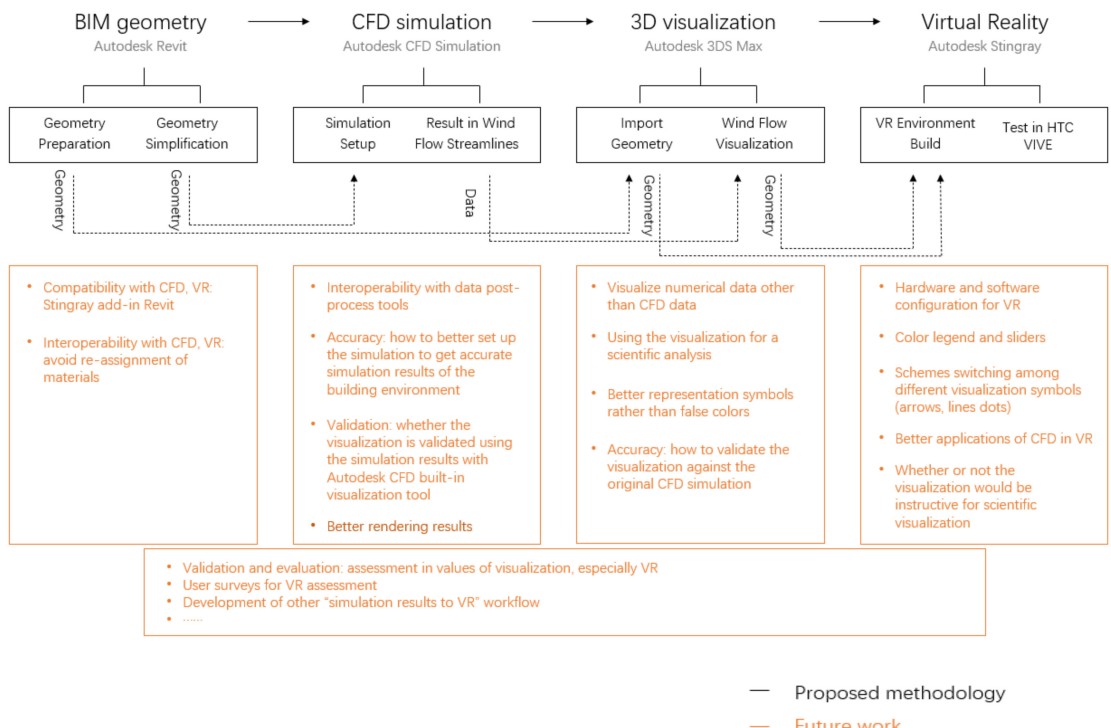

**Figure 18.** Possible future works motivated by this study.

From the aspect of BIM, because a BIM tool was used as the software for providing building information for the later simulation and visualization, the issues pertaining to compatibility and interoperability need to be considered further, such as avoiding the re-assignment of materials during the BIM geometry phase. Revit geometries cannot be directly used in Stingray and 3ds Max, as they require intermediate software; custom scripts might solve this problem or a different set of software programs. New game software programs available with VR capabilities were not available when this case study was created that could help in achieving better interoperability.

Autodesk CFD, as a simulation software, can generate powerful simulation data that could be applied for other use. Post-processing of the data and increasing the interoperability with other software can be further discussed. From the perspective of graphics, the rendering of the color was limited; the false-color display of the data, although scientifically perfect, was not graphically sound in the Autodesk CFD. The developed workflow emphasized the visualization features. There is scope to improve the reliability and accuracy of the final data sent for visualization. The consistency and convenience of Revit to CFD cannot be overlooked.

With regard to 3ds Max, more research should be conducted on the usefulness of this tool to visualize data other than CFD data. Moreover, one should check whether other 3D graphics software would be more user-friendly or fare better in terms of the list of criteria considered. Although the CFD data were represented in 3ds Max, the data cannot be manipulated for data analysis, such as to view data points within a certain range of values. Resolving this issue could be helpful for scientific analyses. Better and more creative graphics for visualization can be developed in future work. As for the methodology, the CFD animation material shader was created for splines to generate a different graphic representation of the splines, in order to help with scientific analysis as well as graphic design.

As VR equipment requires a powerful computer and as VR software has not been fully developed from an architectural perspective, research on the hardware and software for VR, particularly for the building industry, should be conducted. The quality of the graphics is another topic for future research. This could start with a comparison of the visualization based on the number of colors, pixels,

and re-draw time in different VR environments (both software and hardware), and user surveys on what constitutes poor, good, and excellent color/resolution/speed. In the VR environment, color legends can be added to help users to understand the visualization. Interactive sliders that can change the Red Amount would be informative. The question remains in whether the visualization in VR would be useful for actual scientific analysis or more for representation. For more interactions in the VR environment, scheme switching functions between different visualizations (arrow, lines, dots, etc.) can be added. Moreover, interactions, such as moving behaviors, sound, and gestures, are useful elements that can be incorporated into VR. Relevant topics may include the interaction between users and the building environment to better aid in the architectural and engineering design. The airflow effects generated by the CFD simulation should have more applications than just for airflow representation. For example, a CFD simulation could be used for game design, providing actual wind effects. CFD visualization in VR may also be useful in examining hazardous areas within a given space in the presence of toxic gas. People can better understand how to avoid danger. VR nausea, particularly in the presence of moving objects such as moving airflow in the CFD visualization environment, is a topic worth discussing.

The methodology developed in this study was focused on the use of VR as a visualization method for CFD data (and originally weather and solar radiation data). This is just a small part of the whole lifecycle of a building project. In an actual building project, the BIM geometry can be retrieved from conceptual and detailed design stages. Even if in the proposed method the geometry simplification came after the detailed geometry, the model in the conceptual design stage was sufficiently informative for a CFD simulation. Undoubtedly, CFD simulation should belong in the analysis stage. During this stage, different types of simulations, including CFD, were performed to increase the building performance and thermal comfort. The 3D visualization and VR parts were overlapped in the analysis, documentation, and operation/maintenance stages. The reasons are, on one hand, the VR visualization is part of the CFD simulation in different representation methods; on the other hand, the VR result can be documented as part of the deliverables as well as a visualized method to maintain the building after construction.

## 7. Conclusions

Starting with a building information model, architects and engineers can visualize CFD simulation results in a VR environment.

Revit objects (BIM) and airflow information (Autodesk CFD) were implemented through 3ds Max (visualization) and Stingray (VR environment) as an experiment to provide a better representation method for CFD data. Although there are barriers between Revit objects and the VR environment, 3ds Max and other intermediate software programs helped complete the workflow. Using VR to realize architectural design was found to be feasible, and the visualization of the CFD simulation in VR offered more details for architects to comprehend the actual space, thus providing greater assistance in the design process.

From the perspective of the client and designer (architects and engineers), VR could better explain the airflow in a given space, which might also help refine the design. Especially with building environment simulations, clients could give suggestions from their perspective when providing more details, before the actual construction is made. Unnecessary investment can be avoided owing to VR. More communication will take place as clients better understand the design.

The visualization of CFD results in VR has an educational function. For students without any knowledge of building environments, a VR environment can stimulate their interest in learning what the graphics represent, and the exploration of the environment would aid teaching. For students with a building environment background, a detailed virtual environment including interactions will help them to more effectively resolve the possible problems.

Interactive VR has tremendous unrealized potential as another step in the sequence from 2D drafting to 3D modeling to 4D animation to VR in the building industry. As both hardware and

software continue to develop, one can imagine the gaming industry fueling these potentials and hopefully architects and engineers taking advantage of the spin-offs.

Research on CFD visualization in a VR environment using BIM tools was conducted as an experiment to combine several popular technologies that contribute to the development of the building industry in this fast-developing world. BIM tools represent powerful techniques to help with projects during their entire life cycle; newer elements can be added to the conventional building analysis, such as CFD, to play a better role in assisting the architectural design; and the potential use of VR for both architectural visualization and scientific analysis should not be overlooked.

**Author Contributions:** Conceptualization, J.Y. and K.K. (Karen Kensek); methodology, J.Y.; software, J.Y.; validation, J.Y. and K.K. (Karen Kensek); data curation, J.Y.; writing—original draft preparation, J.Y.; writing—review and editing, K.K. (Karen Kensek), D.N. and K.K. (Kyle Konis); visualization, J.Y.; supervision, K.K. (Karen Kensek), D.N. and K.K. (Kyle Konis) All authors have read and agreed to the published version of the manuscript.

**Funding:** This research received no external funding.

**Conflicts of Interest:** The authors declare no conflict of interest.

## Appendix A

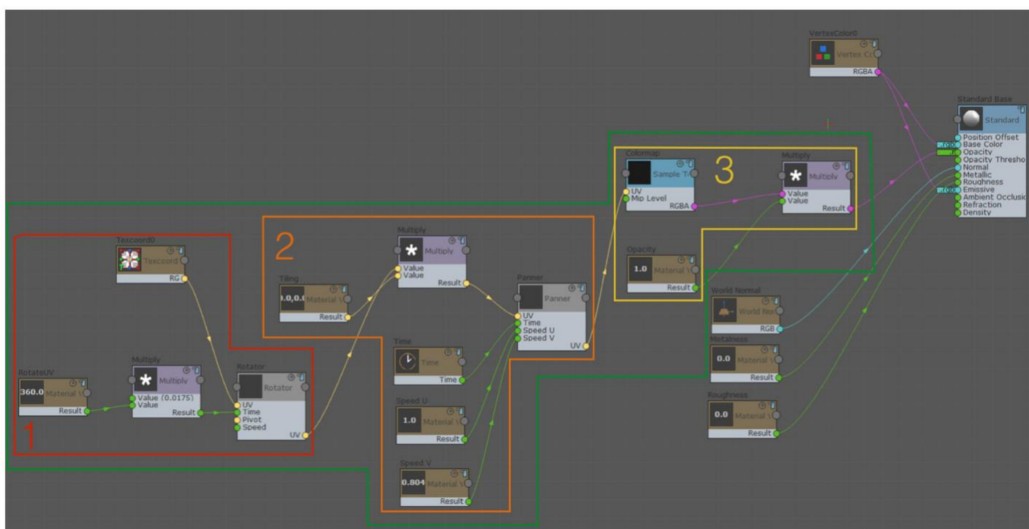

**Figure A1.** CFD animation material script in 3ds Max for wind flow animation.

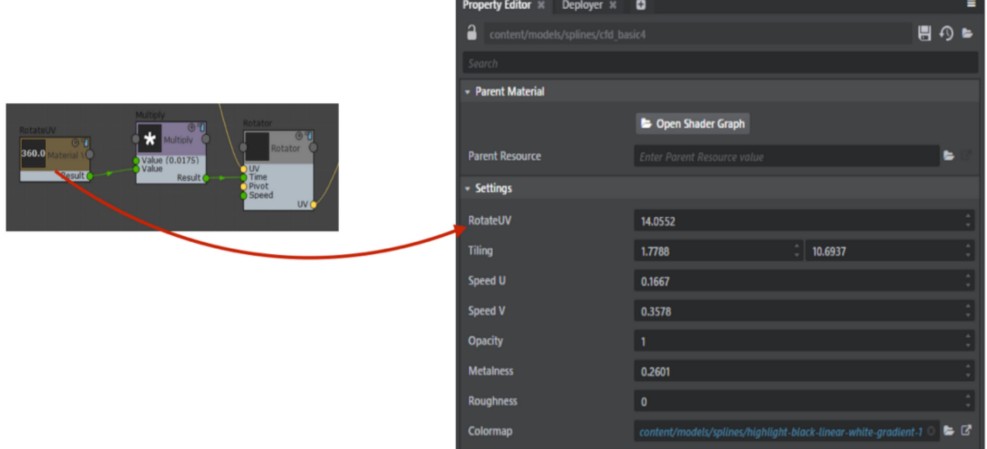

**Figure A2.** Step 1 rotator in the CFD animation material.

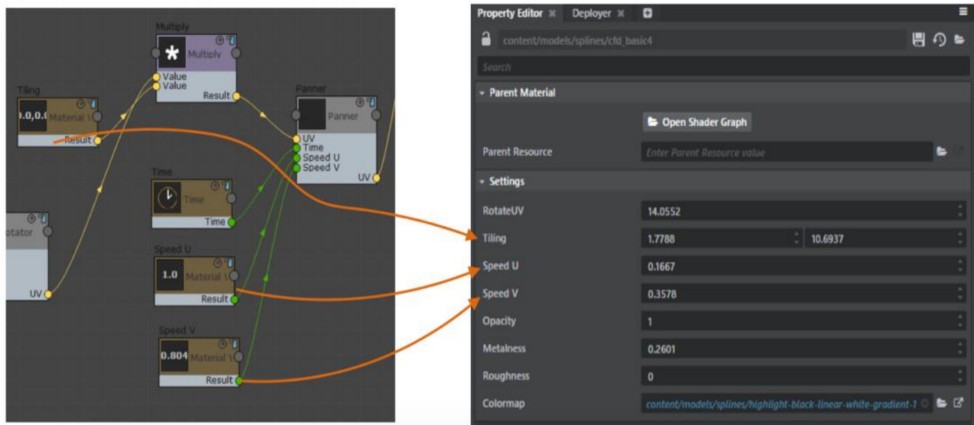

**Figure A3.** Step 2 in CFD animation material.

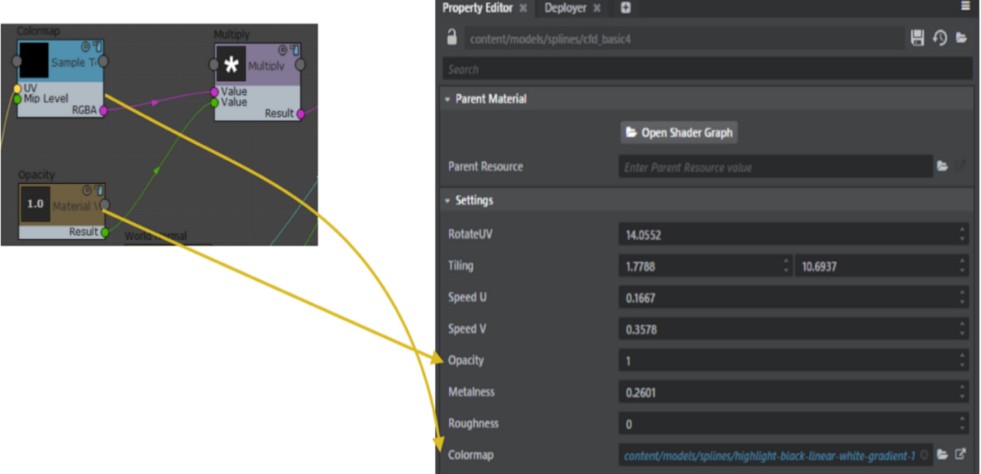

**Figure A4.** Step 3 in CFD animation material.

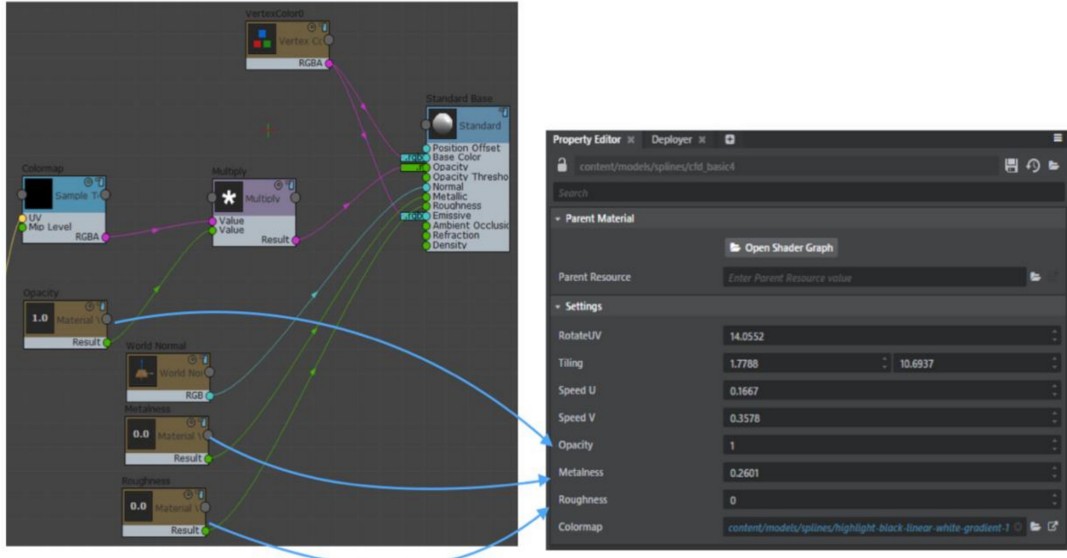

**Figure A5.** Final plug-in in CFD animation material.

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
