# Peer review of "CFD Visualization in a Virtual Reality Environment Using Building Information Modeling Tools"

_buildings, doi:10.3390/buildings10120229_

Round 1

Reviewer 1 Report

Reviewed paper is devoted to the 3D visualizations of computational fluid dynamics (CFD) in the scope of virtual reality (VR) technologies. In general, this is actual and interesting topic. I would recommend following the classical structure of the scientific papers (introduction – materials and methods – results – discussion – conclusion). It would be appropriate to do also other modifications, as described below.

I recommend the paper for acceptance after major revisions.

Introduction

  • Row 51: Previous studies have demonstrated … Which studies?
  • Also, other information given in the introduction deserves to be supported by other relevant and current citations, e.g. lines 87 – 88.
  • There are more studies related to the main topic of this paper, e.g.: Lin et al., 2019, https://doi.org/10.1016/j.autcon.2019.02.007; Zhu et al., 2019, https://doi.org/10.3390/technologies8010004.
  • There are also certain problems and limits associated with VR. These aspects together with the advantages and possibilities of VR are described, for example, by Çöltekin et al., 2020, https://doi.org/10.3390/ijgi9070439.
  • At the end of the introduction, I would expect the list of the main aims of the paper.

Methodology

  • Rows 155-157: The authors use Autodesk software. This selection should be justified. And to emphasize especially why free software was not used (i.e., https://www.paraview.org).
  • This part should describe not only the methodology of creating visualizations but also the methodology of the evaluation.

Case study

  • In my opinion, the whole process is too tied to software from one company.
  • I would recommend generalizing the whole procedure a bit and I would also move the specific examples and information closely related to working in specific software to an appendix (e.g. Figures 13 – 17).
  • Row 237: Source “Liu et al., 2016” is not listed in the References.

Evaluation

  • The description of the evaluation is very vague. Was user testing or expert evaluation used? Further information on evaluation methodology should be provided.
  • How were the percentages shown in Figure 23 determined?

Conclusion

  • The conclusion is quite short, but it is relatively well written.

Future work

  • Potential issues for future work are described too generally. I would recommend moving the vast majority of this section to the discussion, and the last paragraph to the conclusion.

Other minor comments

  • The formatting of citations in the text is unusual, i.e. … [1](Chung et al. 2010) …. on row 21.
  • Check the formatting of citations [1] and [17] in the References list.

Author Response

Dear Reviewer,

Thank you very much for the very constructive feedback on the manuscript. They are very useful for me to get my paper improved and also very instructive for my future researches. 

Overall

The structure of the manuscript has been changed (introduction - literature review - methodology - implementation - results - discussion - conclusion). In this structure I found it would be more clear to state the problem I would like to address.

Introduction

The previous introduction part has been divide into introduction and literature review parts and most of the content have been re-written. The introduction is mainly for why the topic was researched and the literature review was focused on the current researches and what can be learned from them.

The three papers have been studied, which brought a lot of great insights on the focused topic. They have been referenced in the manuscript. 

Major novelty of the paper have been mentioned at the end of the literature review summary. 

Methodology

This part was revised. The software selection was mentioned to explain why major Autodesk software were used. (eg why ParaView was not used question was explained in 2.1 and 2.2 parts as well).

The evaluation method was included in this part. 

Implementation

The case study was an implementation of the proposed methodology and part of the result part was moved and discussed to next part as results. Detailed software instruction figures were moved to appendix.

Results

The implementation results were talked about in this part. The results were discussed in three parts as the visualization results appeared in three software of three phases from the workflow. In this way, it's more reasonable to introduce the evaluation part.

Evaluation was moved to this section. The method used (user feedback test) was mentioned in methodology part

Discussion

The previous future work part was moved here to discuss about the researched topic. 

Conclusion

The paragraph from future work was added in this part

Thank you very much!

Best regards,

Reviewer 2 Report

This paper deals with a meaningful topic called CFD visualization in a virtual reality environment using BIM.

However, from an academic point of view, this paper is reviewed to be very insufficient to consider publication.

First of all, sufficient background on the subject was not provided.

Most of the references provided are not current research literature.

Also, the latest research literature on major keywords such as BIM, CFD, and VR is not provided at all.

For this reason, it is not clear what role BIM can play for CFD visualization in a virtual reality environment.

(line 485) Make sure to properly check the notation of references and spelling errors.

The methodology does not clearly describe the flow of information.

Figure 2 and Figure 24 only show compatibility between commercial software.

In addition, this study raises many questions as follows, but no clear answer can be found in the text.

-It is not known whether the information extracted from BIM geometry should flow sequentially to CFD simulation, 3D visualization, and virtual reality.

-In other words, does 3D visualization be realized only through CFD simulation?

-Is virtual reality realized only through 3D visualization?

-What is the significance of the research using the compatibility of Autodesk's commercial software Revit, CFD simulation, 3ds MAX, and Stingray?

-(line 336: Evaluation) accurracy, graphics, experience Where did the 3 evaluation criteria come from?

-Since the necessary literature review was not provided in the academic paper, many contents are only looked as technical documents for Autodesk software.

Author Response

Dear Reviewer,

Thank you very much for the feedback. They are very constructive for me to improve my work. Major changes have been made as following:

  • The structure of this paper has been changed to 'Introduction-Literature review-Methodology-Implementation-Results-Discussion and future work-Conclusion). In this structure it would provide more background research that support the values of this topic instead of providing technical instructions. 
  • The previous Introduction part has been divided into two part (Introduction and Literature review). State of the art of CFD simulation and visualization, VR, BIM topics were discussed. Most recent researches were included in this part. 
  • In the revised methodology, software selection and data flow  was discussed in part 2.2 and 2.3. The evaluation method was mentioned in 2.4. 

Thank you very much!

Best regards,

Reviewer 3 Report

The authors propose a methodology to incorporate Building Information Modelling (BIM) tools to visualize and even manipulate computational fluid dynamics (CFD) data using virtual reality (VR) to study the life cycle of a building. For that, they use BIM geometry, CFD simulation, 3D and VR visualization with Autodesk software (Revit, CFD, 3ds Max and Stingray) to facilitate data exchange and compatibility. The authors apply this methodology to kitchen space in a family house, as a case study. They compare the visualization with different software (CFD, 3ds Max and Stingray) in terms of accuracy, graphics and experience.
In the opinion of this reviewer, it does not appear clear enough if the authors only want to compare the different software graphically and not to study the airflow in the case study or not. For example, it would be interesting to incorporate in the case study the windows for natural ventilation and the smoke extractor in the stove to see the interferences with the airflow from the inlet ducts to the outlet ducts.
At the end of the First Section, together with the three main conclusions, the authors may include the main novelty of the research and its main application to buildings lifecycle. In Section 5, they may also expose more accurately the conclusions of the study and not just the importance of the representation of the CFD data. 

Author Response

Dear Reviewer,

Thank you very much for the recognition of the manuscript. The manuscript has been revised and improved as to provide a more thorough look of the research.

This research offers a method for CFD visualization in VR using BIM. It may be insufficient to talk about the airflow or ventilation in the simulated area. It would be an valuable topic in the future work. 

The novelties and aims have been concluded briefly after the literature review. And the Results and Conclusion parts have been improved.

Thank you very much!

Best regards,

Round 2

Reviewer 1 Report

Most of my suggestions have been responded in the text of the paper. The clarity of the article has increased. I have now only a few following comments. I consider most serious the comment n. 2.

  1. Check that all abbreviations are explained at the first occurrence in the text (e.g. AR, VR, XR, MEP).
  2. The description of the evaluation is still a bit vague. Did the “8 random users” have any knowledge of the domain (architecture, interior design, construction, etc.)? Instead of 37.5%, it is better to state the absolute number of participants (i.e. 3 out of 8), when the total number of users was so low. How exactly did the rating work? Did users choose from scale poor – good – excellent?
  3. Schema “Case study visualization results evaluation & comparison” (page 20) is not marked as a figure or described. By the way, this title is a bit awkward.
  4. Readability of Fig. 17 (labels in graphs) is not ideal, but this can be a problem for automatic PDF generation in the content management system.

Author Response

Dear Reviewer,

Thank you for the comments and suggestions. They helped a a lot for me to better clarify my work.

  1. Abbreviations have been checked. I have added necessary explanations for them: 1) AR and MR: row 100, 2) VR: row 4, 3) XR: row 99, 4) MEP: row 323
  2. It is more appropriate to use numbers instead of percentage to describe the user test results. I have removed made the change in section 5.6. And I have added more details in section 2.4 and 5.6 to explain the rating method and results.
  3. The figure "Case study visualization results evaluation and comparison"  appeared in the very first manuscript. It was removed as it might be a little bit redundant. The updated Figure 17 has been placed on page 15.
  4. I have edited Figure 17 to make it more clear in the PDF version. 

Thank you very much for your guidance on this work!

Best regards,

Jiayi Yan

Reviewer 2 Report

This manuscript is greatly improved over the previous version.

In the previous review phase, my comments were similar to other reviewers.

Fortunately, my comments have been faithfully reflected in the revised manuscript.

As it is, it looks good enough to publish on 'Buildings'.

Thank you for review request.

Author Response

Dear Reviewer,

Thank you very much for your comments and suggestions. I have learned a lot from them as to improve this work as well as other of my research. I am very glad to have your approval.

Thank you!

Best regards,

Jiayi Yan